# The Role of Obesity in Type 2 Diabetes Mellitus—An Overview

**DOI:** 10.3390/ijms25031882

**Published:** 2024-02-04

**Authors:** Preethi Chandrasekaran, Ralf Weiskirchen

**Affiliations:** 1UT Southwestern Medical Center Dallas, 5323 Harry Hines Blvd. ND10.504, Dallas, TX 75390-9014, USA; 2Institute of Molecular Pathobiochemistry, Experimental Gene Therapy and Clinical Chemistry (IFMPEGKC), Rheinisch-Westfälische Technische Hochschule (RWTH), University Hospital Aachen, D-52074 Aachen, Germany

**Keywords:** obesity, type 2 diabetes, pathophysiology, incidence, prevalence, management, therapeutic approach, in vivo studies, clinical trials

## Abstract

Obesity or excessive weight gain is identified as the most important and significant risk factor in the development and progression of type 2 diabetes mellitus (DM) in all age groups. It has reached pandemic dimensions, making the treatment of obesity crucial in the prevention and management of type 2 DM worldwide. Multiple clinical studies have demonstrated that moderate and sustained weight loss can improve blood glucose levels, insulin action and reduce the need for diabetic medications. A combined approach of diet, exercise and lifestyle modifications can successfully reduce obesity and subsequently ameliorate the ill effects and deadly complications of DM. This approach also helps largely in the prevention, control and remission of DM. Obesity and DM are chronic diseases that are increasing globally, requiring new approaches to manage and prevent diabetes in obese individuals. Therefore, it is essential to understand the mechanistic link between the two and design a comprehensive approach to increase life expectancy and improve the quality of life in patients with type 2 DM and obesity. This literature review provides explicit information on the clinical definitions of obesity and type 2 DM, the incidence and prevalence of type 2 DM in obese individuals, the indispensable role of obesity in the pathophysiology of type 2 DM and their mechanistic link. It also discusses clinical studies and outlines the recent management approaches for the treatment of these associated conditions. Additionally, in vivo studies on obesity and type 2 DM are discussed here as they pave the way for more rigorous development of therapeutic approaches.

## 1. Introduction

The rapid development of global urbanization and modernization has lasting effects on lifestyle aspects such as unhealthy eating habits, lack of exercise, increased stress and environmental factors. These factors contribute to the alarming growth of obesity and type 2 DM worldwide. Obese individuals develop insulin resistance, which is characterized by impaired insulin action in the liver and reduced glucose uptake in fat and muscle [1]. While lifestyle changes and medications are recommended for prevention, they have not been successful in suppressing the increasing incidence conditions. Therefore, it is crucial to gain a deeper understanding of the molecular mechanisms linking obesity and type 2 DM in order to address this global healthcare challenge effectively.

The intricate connections and sharing of pathophysiological mechanisms between obesity and type 2 DM amplify the prevalence and incidence of insulin resistance, dyslipidemia, NAFLD and a constellation of metabolic abnormalities in obese individuals. Increased body mass index (BMI) and abdominal fat distribution linearly increases the risk of type 2 DM due to alterations in adipose tissue biology that links obesity with insulin resistance and beta cell dysfunction [1]. Abdominal obesity, commonly determined by waist-to-hip ratio, is an independent factor for the development of hypertension and elevated fasting glucose, even if the overweight individual with predominant abdominal fat does not meet the BMI criteria for obesity [2]. Multiple in vivo and clinical studies have demonstrated a cause-and-effect relationship between obesity and type 2 DM, unraveling their intimate connections. It is shocking to note that according to the WHO fact sheet, at least 41 million children under the age 5 are overweight or obese (BMI ≥ 35 kg/m^2^) as of 2016. If this trend continues, 60% of the world’s population will be obese or overweight by 2030 [3]. According to WHO, obesity accounts for 44% of diabetes and the incidence of obesity-related diabetes is expected to double to 300 million by 2025 [4].

It is interesting to note that during starvation, adipose tissue, which serves as a major fuel reserve, provides a critical energy source for survival. Additionally, adipose tissue modifies various physiological functions, including appetite, reproduction and insulin action, through the secretion of adipokines and exosomes [1]. In obese individuals, non-esterified fatty acids play a crucial role in the development of insulin resistance and beta cell dysfunction [1].

The close relationship between obesity and diabetes has led to the term “diabesity”, which highlights that the majority of individuals with diabetes are obese or overweight [5]. While type 2 DM is influenced by genetic predisposition and ethnicity which are non-modifiable risk factors, it can still be prevented or managed by addressing modifiable risk factors such as obesity. Despite recent advancement in management strategies, obesity and diabetes remain a significant interconnected public health challenge worldwide. In this review, we will explore the mechanistic link between obesity and diabetes, discuss clinical and in vivo studies that focus on this association and briefly touch upon the clinical definitions of obesity, diabetes and epidemiology.

## 2. Definitions and Epidemiology of Obesity and Type 2 DM

Obesity is defined as the excessive accumulation of fat in various parts of the body or organs, known as ectopic fat or throughout the body. It is a chronic, progressive, relapsing condition with multiple factors that lead to adverse metabolic and psychosocial health consequences [6]. One of the main causes of obesity is an imbalance between the excess energy stored and the energy utilized by the body, which can disrupt nutrient signals and result in insufficient energy expenditure [6]. Assessing the risk factors for adiposity involves measuring height, weight, BMI, waist circumference and body fat percentage. The diagnosis of obesity relies on the BMI cut off and the relationship between body weight, fat distribution pattern and visceral fat [7]. BMI alone is no longer sufficient to evaluate obesity, as it is a diverse condition. Table 1 displays the classification of obesity based on BMI and waist circumference [7].

The anthropometric assessment of percentage body fat is a more accurate measure of adiposity than BMI [8]. The Obesity Medicine Association classification of percentage body fat is shown in Table 2 [8].

It is interesting to note that by 2030, an estimated 14% of men and 20% of women in the world’s total population will develop clinical obesity. Additionally, it is estimated that 18% of individuals will have a BMI greater 30 kg/m^2^, 6% will have a BMI greater than 35 kg/m^2^, and 2% will have a BMI greater than 40 kg/m^2^ [9]. According to the World Obesity Federation, countries with high socioeconomic status and per capita income are at a greater risk of experiencing an increased prevalence of obesity [9]. It is alarming to note that approximately two-thirds of the adult population in the United States is either obese or overweight [10].

The prevalence of obesity in the United States has alarmingly increased over the past decade. According to published data from 2017 to 2020, 42.4% of adults have a BMI ≥ 30 kg/m^2^, while 20.9% of youth have a BMI ≥ 30 kg/m^2^. Additionally, the age-adjusted prevalence of severe obesity, defined as BMI ≥ 40 kg/m^2^, is 9.2% [11].

Currently, only around 30% of the adult population in the US has a normal BMI between 18–25 kg/m^2^ [12]. When considering race and ethnicity, the highest rates of obesity are found among black women, Native Americans and Hispanics [13]. It is projected that by 2023, approximately 50% of the US adult population will be obese with around 25% developing severe obesity [14].

From a global perspective, the obese population worldwide has risen steadily over the past few decades with a six-fold increase in adults with obesity between 1975 and 2016. This increase is driven by socioeconomic advances, which pose a significant healthcare burden and contribute to the rise in mortality complications associated with obesity, such as DM and cardiovascular diseases [15].

Obesity plays an inevitable role in the increased prevalence of type 2 DM, a chronic condition where the body fails to produce sufficient insulin or cannot efficiently utilize insulin resulting in elevated blood glucose levels as its primary manifestation. Type 2 DM is characterized by low insulin secretions from β-cells and peripheral insulin resistance, leading to elevated levels of fatty acids. This causes a decrease in glucose transport into muscle cells, increased fat breakdown and hepatic glucose production [16].

This is the fastest-growing pandemic and health emergency globally. According to the latest estimates by the International Diabetes Federation, the number of diagnosed cases of DM is predicted to reach 643 million by 2030 and 783 million by 2045 [17]. Additionally, the majority of undiagnosed type 2 DM cases are concentrated in Africa, Southeast Asia and the Western Pacific. In 2021, there were 541 million adults diagnosed with impaired glucose tolerance and 319 million adults with impaired fasting glucose. These numbers are predicted to increase to 730 million and 441 million, respectively, by 2045 [17].

Previous studies have shown that the relative risk of developing type 2 DM is 4.6-fold higher for woman and 3.5-fold higher for men with a BMI greater than 29.9 kg/m^2^ compared to their same-sex peers with a BMI less than 24.9 kg/m^2^ [18]. It is important to note that the associations between central obesity and comorbidities vary among different races and ethnicities (Figure 1).

For instance, in Asian populations with type 2 DM, central obesity is a more accurate predictor than BMI [20]. Interestingly, for individuals of European ancestry, the thresholds for central obesity are determined by a waist circumference greater than 94 cm (37 inches) for men and greater than 80 cm (31.5 inches) for women. These thresholds for individuals of South Asian, Chinese and Japanese origin are greater than 90 cm (35.5 inches) for men and greater than 80 cm (31.5 inches) for women [21,22].

Diabetes is a chronic metabolic disorder with multiple causes, characterized by consistently high blood glucose levels due to defects in insulin secretion, action or both. Type 2 DM is more common than type 1 DM, accounting for 90–95% cases. It is strongly influenced by genetics and involves resistance to insulin action and inadequate compensatory insulin secretion [23]. Most patients with type 2 DM are obese, with a higher percentage of body fat or abnormal distribution, which is related to the pathophysiology of DM. Adipose tissue promotes insulin resistance by releasing more free fatty acids [24]. Other contributing factors include peripheral insulin resistance, dysregulation of hepatic glucose production, decreased beta cell function and beta cell failure [1].

The diagnosis of type 2 DM is made when the patient meets one of the following criteria: glycated hemoglobin (HbA1_C_) ≥ 6.5%, fasting blood glucose ≥ 126 mg/dL or 2-h post-prandial glucose ≥ 200 mg/dL. Diabetes-related morbidity and complications can be substantially reduced with tight glycemic control, aiming for an HbA1c of less than 7% [25].

According to the International Diabetes Federation (IDF), 415 million adults aged 20–79 years were diagnosed with DM in 2015 and the number of people suffering from diabetes increase in that group rose to about 573 million adults in 2021 (Figure 2).

The global population with DM is predicted to increase to another 200 million by 2040 [27]. In the United States, Native Americans, Hispanics and Asian Americans are the most affected population by type 2 DM [28]. It is estimated that 70% of individuals with pre-diabetes will eventually develop type 2 DM by 2030 [27].

Interestingly, certain regions such as Fiji and American Samoa have reported the highest prevalence of the disease. Southeast Asian countries have also seen an increase in the last two decades. The top spots with the greatest total number of individuals with type 2 DM include China (88.5 million), India (65.9 million) and the US (28.9 million), which can be attributed to their large population size [29].

## 3. Mechanistic Link

### 3.1. White and Brown Adipose Tissue

It is crucial to understand the difference between classifications and subsets of adipose tissue. Morphologically, adipose tissue can be classified as white adipose tissue (WAT) or brown adipose tissue (BAT). WAT can be further divided into two subsets: subcutaneous adipose tissue and visceral adipose tissue. Subcutaneous adipose tissue is characterized by small adipocytes that exhibit stronger anti-lipolytic effects of insulin. On the other hand, visceral adipose tissue is characterized by larger adipocytes and is metabolically very active. In healthy lean individuals, the WAT mass is confined to specific depots. However, in cases of obesity, the WAT mass increases ectopically in areas such as mesenteric, omental or retroperitoneal fat [30,31].

Brown fat, which comprises 1–2% of total fat stores, functions by generating heat through the uncoupling of the respiratory chain. This allows for the fast production of ATP through oxidative phosphorylation. This process helps clear plasma triglycerides and prevents the storage of excess fat in other areas of the body. Brown fat also acts as a reservoir for excess blood glucose and free fatty acids, aiding their disposal. Non-shivering thermogenesis, mediated by UCP1 in BAT, is considered a promising way for increasing energy expenditure. The differentiation of resident progenitor cells into mature adipocytes is facilitated by critical factors such as BMP7 and EBF2, leading to the development of a beige pre-adipocyte phenotype. This is followed by the transcriptional activation of UCP-1, PRDM16 and Zfp516. A crucial step in this differentiation process involves the deacetylation of PPARγ by SIRT1 and the promotion of mitochondrial biogenesis. Additionally, the browning machinery also involves PGC-1α and C/EBPs [32].

Numerous studies have shown that dysfunction in BAT contributes to insulin resistance and hyperlipidemia. Multiple rodent studies have demonstrated that transplanted BAT improves glucose tolerance, insulin sensitivity and reduces obesity. Furthermore, it has been found that fibroblast growth factor 21 (FGF21) stimulates the browning of WAT, activating brown adipocytes and inducing thermogenesis. A clinical trial involving LY2405319, a variant of FGF21, in obese individuals with type 2 DM showed improved body weights, decreased triglyceride levels and improved insulin sensitivity [33].

The role of macrophages in eliciting metabolic inflammation and their interactions with adipose cells have been well-known in obesity-associated insulin resistance for years. Interestingly, the number of macrophages in brown fat depots is reported to be low, suggesting that BAT has an inherent ability to reduce the inflammatory capacity of macrophages compared to WAT, enhancing this profile. Therefore, it is evident that the activation of brown adipose tissue in WAT-mediated thermogenic activity counteracts insulin resistance and metabolic dysfunction [34].

### 3.2. Adipogenesis and Healthy Adipose Tissue

PPARγ is the master regulator of adipogenesis and the maintenance of differentiation from pre-adipocytes to mature adipocytes. It is important to note that other factors such as C/EBPs, KLFs, PRDM16 and PGC-1α are also involved in this regulation. PPARγ upregulates the expression of C/EBPα by promoting its transcription, which, in turn, is associated with elevated expression of other adipogenic genes. PRDM16 functions to suppress white adipocyte specific genes by forming complexes with terminal binding proteins CTBP1 and CTBP2. The activation of brown fat-specific genes occurs when CTBPs are displaced through the recruitment of PPARγ co-activators PGC-1α and PGC-1β [35,36].

PGC-1α, a master transcriptional cofactor for mitochondrial biogenesis, is mainly expressed in BAT. One of the main downstream mediators of PGC-1α is the transcription factor A mitochondria (TFAM), which coats and stabilizes individual mtDNA inducing promoter activity. Furthermore, PGC-1α is involved in the response to oxidative stress by inducting SIRT3, which plays a critical role in β-oxidation and anti-oxidative reactions by modulating acetylation levels of mitochondrial enzymes. The crosstalk between adipogenesis and mitochondrial biogenesis confirms the involvement of PGC-1α in obesity-linked insulin resistance, as it is widely accepted as a marker of transdifferentiation of white into brown adipocytes [37,38].

Recently, multiple studies have shed light on the central role of HO-1 in maintaining beige-like adipose tissue and improving liver functions and insulin sensitivity. HO-1 protects against obesity-induced insulin resistance through the degradation of pro-oxidant heme and the production of carbon monoxide and bilirubin, which have anti-inflammatory properties. As a result of obesity, the levels of reactive oxygen species (ROS) within the adipocytes increase, leading to repression of HO-1 and SOD. This, in turn, increases the production of pro-inflammatory cytokines. Recent research has shown that HO-1 in adipocytes can reverse the detrimental effects of obesity, including insulin resistance and dyslipidemia [39,40].

### 3.3. Dysfunctional Adipogenesis

Adipose tissue is the major reservoir for storing or releasing lipids, depending on fuel availability. Impairment in the pathways associated with the differentiation and proliferation of precursor cells into adipose cells leads to impaired adipogenesis and reduced ability to store excess lipids. This subsequently leads to insulin resistance due to the accumulation of ectopic fat. Multiple studies suggest that BMP4 signaling is important for the recruitment and differentiation of adipocytes and the development of a brown phenotype, which protects against obesity and obesity-linked insulin resistance. Dysregulation of BMP4 signaling can lead to adipocyte hypertrophy and systemic metabolic dysfunction [41].

The multi-dimensional contributions to insulin resistance such as specific nutrients, growth factors, a high-calorie diet and incretin, play a significant role in the development of type 2 DM (Figure 3) [42].

### 3.4. Adipose Tissue Dysfunction and Inflammation

As previously mentioned, adipose tissue is highly flexible and can adapt to rapid changes in energy balance during periods of fasting and feeding. However, if this adaptive response is altered, it can lead to the development of metabolic dysfunction in people with obesity [43]. Numerous in vivo studies have revealed a variety of complex biological and physiological processes in adipose tissue that contribute to insulin resistance. In particular, adipose tissue produces numerous adipokines such as adiponectin, leptin, visfatin, resistin, apelin, omentin, retinol binding protein (RBP4), vaspin and many others that impact the overall activity of different organs including the liver, pancreas, gut, brain and skeletal muscle (Figure 4).

These processes include adipocyte hypoxia, which is caused by increased oxygen demand and stimulates fibrogenesis and macrophage chemotaxis. This leads to an increase in branched chain amino acids concentration in the plasma, an increase in the number of adipocyte macrophages and T cells, a decrease in adipose tissue production and adiponectin (an insulin sensitizing hormone), an increase in lipolytic activity of adipose tissue leading to the release of free fatty acids into circulation and alterations of exosomes derived from adipose tissue macrophages [44,45,46,47,48]. Among these factors, it has been proposed that adipose tissue inflammation is the main driving force behind insulin resistance in obese individuals [49]. Adipocyte hypoxia occurs because of the activation of saturated FFA stimulated adenine nucleotide translocase 2 (ANT2), an inner mitochondrial protein that is triggered by a high fat diet and obesity.

This exacerbates dysfunction and inflammation in adipose tissue, causing activated macrophages and hypertrophied adipocytes to increase the levels of pro-inflammatory cytokines such as TNF-α, IL-1β, monocyte chemoattractant protein-1 (MCP-1) and IL-6, which ultimately leads to the development of metabolic inflammation [50]. This chronic inflammatory state is a key factor contributing to the pathogenesis of insulin resistance, which decreases the glucose uptake in muscle, leads to increase glucose production in liver, provokes β-cell dysfunction in pancreas and results in endocrine dysfunction of adipose tissue (Figure 5).

Adipose tissue macrophages also play a causative role in obesity-associated insulin resistance. Recent studies have shown that decreased macrophage recruitment in obesity can alleviate insulin resistance in animal models [51].

Few studies have deciphered the role of toll-like receptors in inflammation-associated insulin resistance in obesity. For example, one study revealed that the abolition of TLR4 alleviated obesity-induced insulin resistance through reducing oxidative stress by metabolic reprogramming of mitochondria in visceral fat [52].

The signaling pathways involved in the inflammatory mechanisms include the activation of JNK1 by TNF-α, which lead to serine phosphorylation of insulin receptor substrate 1 (IRS1) and impairs the action of insulin [53]. Other inflammatory mediators attributed to obesity-induced insulin resistance belong to the class of suppressor of cytokine signaling (SOCS) proteins. These proteins function as part of a feedback pathway in cytokine signaling. The inhibition of insulin signaling by cytokines occurs through interference with the tyrosine phosphorylation of either IRS1 or IRS2 [54].

It is well known that adipose tissue maintains whole-body energy homeostasis. Obesity, which occurs due to the accumulation of WAT in visceral organs, leads to a lack of angiogenesis in subcutaneous adipose tissue. This results in an inability to store excess energy, leading to insulin resistance in obese individuals. Therefore, there may be a connection between adipose tissue angiogenesis, vascular function and insulin sensitivity. A study supported this idea by showing improved metabolic homeostasis in mice when capillary-derived beige adipocytes were implanted [55].

### 3.5. Adiponectin

An important biomarker of adipose tissue health is plasma adiponectin, which is directly associated with insulin sensitivity and has an inverse relationship with the percentage of body fat [56]. Adiponectin has a variety of functions, including anti-inflammatory, anti-fibrotic and insulin-sensitizing effects. These effects are partly mediated by increased ceramidase activity and decreasing intracellular ceramide levels. Adiponectin levels are low in obesity [57]. In the liver, adiponectin decreases the influx of fatty acids, increases fatty acid oxidation and reduces hepatic glucose output. In the muscle, adiponectin stimulates fatty acid oxidation through AMP-activated protein kinase. Although the release of free fatty acids from visceral fat was previously believed to cause insulin resistance, it is unlikely to be a major factor since only 20% of the free fatty acids are derived from the lipolysis of visceral fat [58].

### 3.6. Other Adipokines and Cortisol

Plasminogen activator inhibitor-1 (PAI-1), a member of the serine protease inhibitor family, is elevated in obesity and insulin resistance and predicts the risk of type 2 DM [59]. IL-6 levels predict the development of DM, and when administered peripherally in mice, Il-6 induces hyperglycemia, hyperlipidemia and insulin resistance by the downregulating of IRS and the upregulating of SOCS-3 [60]. Cortisol causes insulin resistance and type 2 DM by opposing the anti-gluconeogenic effects of insulin in the liver [61].

### 3.7. Lipids and Free Fatty Acids

The contribution of increased gluconeogenesis in obese individuals causes fasting and postprandial hyperglycemia. This is because these individuals have an impaired insulin ability to suppress hepatic glucose production. The increase in hepatic gluconeogenesis is due to impaired suppression of adipose tissue lipolysis, which increases the delivery of free fatty acids to the liver [62]. Defective AKT activation promotes the activation of lipolytic enzymes, which impairs GLUT4 translocation to the membrane and worsens hyperglycemia [63].

In obese individuals, the increased lipid oxidation causes a high concentration of plasma-free fatty acids, which leads to insulin resistance. This ‘lipid overflow’ limits the oxidation of glucose and, as a result, in skeletal muscle, the preference for using free fatty acids as an energy source inhibits glycogen synthase and restricts the use of glucose from glycogen stores. Consequently, there is a compensatory increase in plasma glucose and insulin levels over time. However, despite this compensatory effect, there is still resistance to glucose uptake, resulting in type 2 DM (Figure 6) [64].

Previously, it was hypothesized that the competition between increased circulating fatty acids and glucose for oxidative metabolism in insulin responsive cells explained the association between obesity and type 2 DM [66]. However, more recent research suggests that glucose uptake is the rate-limiting step in fatty acid-induced insulin resistance rather than intracellular glucose metabolism [67]. According to this model, fatty acids, along with metabolites such as acyl-CoA, ceramides and diacylglycerols, serve as signaling molecules that activate protein kinases. These protein kinases impair insulin signaling by increasing the serine phosphorylation of insulin receptor substrate which are key mediators of insulin receptor signaling [67].

### 3.8. Distribution of Fat and Ectopic Fat Storage

A striking feature in determining the risk of adiposity induced type 2 DM is the distribution pattern of fat throughout the body. For example, obese individuals with abdominal subcutaneous, intraabdominal fat, intrahepatic triglycerides and pancreatic fat are at a higher risk of developing type 2 DM compared to people with lower body fat accumulation [68]. The role of intramyocellular lipids (IMCL) is increasingly recognized as an important factor in modulating insulin action. Several transgenic animal models have demonstrated that decreased IMCL improves insulin sensitivity [69]. The accumulation of IMCL in obese patients is due to increased levels of lipid peroxidation and the production of lipid peroxidation byproducts such as 4-hydroxynonenal (4-HNE) as well as decreased muscle fatty acid beta oxidation which affects insulin sensitivity [70].

Ectopic lipid accumulation has been implicated in insulin resistance due to the activation of reactive oxygen species (ROS), mitochondrial dysfunction or endoplasmic reticulum (ER) stress [71]. The persistent imbalance between the production of ROS and antioxidants is the main cause of oxidative stress, leading to fat accumulation in both humans and mice [72]. The increase in ROS in pre-diabetes is caused by an increase in fatty acids which results in oxidative stress due to increased mitochondrial uncoupling and beta oxidation subsequently, leading to increased ROS production [73]. Multiple in vitro studies have elucidated that ROS and oxidative stress activate potential targets of the insulin-signaling pathway such as the insulin receptor and IRS proteins [74]. In insulin resistance, ectopic fat accumulation is associated with decreased mitochondrial oxidative activity and ATP synthesis. The accumulation of intramyocellular fat is attributed to decreases in muscle mitochondria due to decreased expression of PPARγ and PGC-1β, which regulate mitochondrial biogenesis [75].

A possible mechanism for the generation of excess ROS is the excessive formation of superoxide by NADPH oxidase (NOX). This inactivates the critical metabolic enzymes, initiating lipid peroxidation in obesity. The excessive production of ROS activates numerous pro-inflammatory cytokines including NF-κB, which promotes insulin resistance. The dysregulation or ROS causes disturbances in multiple biochemical mechanisms such as oxidative phosphorylation, the generation of superoxide from NOX, PKC activation and hyperleptinemia. Additionally, the activation of serine kinases by ROS results in hyperphosphorylation of serine/threonine, inhibiting the tyrosine phosphorylation of IRS1 and IRS2 thereby diminishing insulin signaling [76].

In the case of lipotoxicity, the normal dynamics of mitochondria are altered, damaging mitochondrial DNA and increasing mitophagy, a process through which dysfunctional mitochondria are removed through lysosome fusion. This increase in mitophagy decreases the number of mitochondria, exacerbating lipid accumulation and lipotoxicity, which causes mitochondria-mediated cell apoptosis. Thus, mitochondrial dysfunction causes an overall alteration of metabolism, increased respiration and excessive acetyl CoA production. It is fascinating to note that the characteristics of mitochondria are also altered in obese states. This includes increased lipid peroxides, increased DAG and ceramides due to incomplete fatty acid oxidation. The combined effect of mitochondrial dysfunction and oxidative stress produces damage-associated molecular patterns that activate inflammatory signaling pathways such as TLR9-neutrophil-NF-κB pathways and NLRP3/Caspase 1/IL-1β pathways [77].

Another pathway implicated in insulin resistance is ER stress. One possibility for how obesity leads to ER stress is that ectopic lipid storage triggers ER stress through mechanical stress or disruptions in intracellular nutrient and energy fluxes, as well as changes in tissue architecture [78]. Strikingly, chronic overnutrition and obesity generate an excess of unfolded protein response transducers, which in turn leads to the release of pro-inflammatory cytokines. Ultimately, this process leads to the development of insulin resistance [79].

### 3.9. Disturbances in Lipid Homeostasis

Dyslipidemia is a hallmark feature of type 2 DM, characterized by increased levels of triglycerides, triglyceride-rich lipoproteins and decreased levels of high-density lipoproteins. Normally, chylomicrons deliver dietary lipids to the liver. Once in circulation, lipoprotein lipase, activated by apolipoprotein C-II, breaks down the triglycerides in the chylomicron core, releasing free fatty acids. As triglycerides are progressively removed, chylomicron remnants are formed and cleared by hepatocytes with the incorporation of apolipoprotein E (ApoE). This, along with the uptake of free fatty acids generated by lipolysis, is the main source of very low-density lipoproteins (VLDL) assembly and secretion. VLDL is gradually broken down to form smaller VLDL particles, and eventually low-density lipoproteins (LDL). However, in type 2 DM and insulin resistance, the efficient process of lipoprotein metabolism and clearance is impaired, leading to disruptions in the clearance of VLDL and chylomicrons [65,80,81,82].

## 4. In Vivo Models

Multiple in vivo studies have been conducted to shed light on the complex pathways that connect obesity and type 2 DM, which have the potential to be translated into clinical studies. A recent study found that PAI-1 levels were higher in obese individuals with insulin resistance compared to obese individuals without insulin resistance [83]. Additionally, a study using rodent models showed that overexpression of PAI-1 in adipocytes led to insulin resistance, while the knockout of PAI-1 in adipocytes improved insulin action [84].

Animal studies have demonstrated the importance of intramyocellular lipids in regulating insulin action. For example, the knockout of the fatty acid translocase CD36 resulted in improved insulin sensitivity and reduced fatty acid uptake, despite increased levels of plasma free fatty acids [85].

In another interesting study, a KK-Ay transgenic mouse model was developed to exhibit hyperglycemia, hyperinsulinemia and moderate obesity. The effect of Exendin 4, a GLP1-1 receptor agonist, was tested and shown to increase insulin secretion and reduce glucose levels [86,87].

The ob/ob mouse model, generated in 1949, is the most popular model for severe hyperglycemia and a monogenic model of obesity caused by a lack of leptin production. This model has been widely used in numerous studies. For instance, treatment with Sibutramine for six weeks in six-week-old ob/ob mice resulted in a 12% reduction in weight gain, decreased plasma insulin levels and improved insulin sensitivity [88,89].

Another commonly used obese rat model is the Zucker rat, in which the regulation of the leptin signal for reduced food intake is non-functional, resulting in preferential deposition of lipids in adipose tissue. By 14 weeks, these Zucker rats have a fat percentage of 40%. Although these animals exhibit insulin resistance and obesity, they are not overtly diabetic [90,91]. The effect of Sibutramine on these rats was examined, demonstrating a significant reduction in food intake [92]. A sub-strain of the Zucker rats, known as the Zucker diabetic fatty rats, has been widely utilized as an animal model to test various anti-diabetic and anti-obesity drugs. For instance, Liraglutide significantly attenuated the progression of diabetes in these rats, along with a reduction in blood glucose levels [93].

An important limitation of monogenic models deficient in leptin is that they do not accurately represent the pathogenesis of obesity in humans. Therefore, mouse and rat models induced by diet were developed to mimic human-like conditions. For instance, the C57BL6/J strain is commonly used to create a diet-induced obese model due to its propensity for obesity [94]. The commonly used rat strains for developing diet-induced obesity (DIO) models are Wistar, Sprague–Dawley, Long Evans, and Osborne Mendel rats [95]. In these DIO rats, Sibutramine and Liraglutide have shown effects on reducing body weight that are comparable to those in humans. However, the effects on plasma lipids and glucose tolerance still need to be further explored [96]. One major drawback of DIO rat models is that they often develop hyperinsulinemia instead of hyperglycemia. This has led to the development of a polygenic rat model that exhibits adult-onset obesity, insulin resistance and type 2 DM without the need for dietary intervention while maintaining leptin signaling [97].

One striking model is the development of UC Davis-type DM rats, in which Liraglutide (0.2 mg/kg) demonstrated a decrease in body weight and delayed or prevented the onset of diabetes. This treatment also led to a decrease in fasting plasma insulin levels and improved insulin sensitivity [98,99].

Another model, the New Zealand obese mouse (NZO) polygenic model crossed with non-obese nondiabetic (NON) mouse, resulted in NONcNZO10/LTJ mice. In these mice, it was shown that the selective β3 adrenergic receptor agonist CL316,243 reduced body weight and suppressed the development of diabetes [99,100].

A polygenic mouse model with moderate obesity and male-derived hyperglycemia is the Tallyho mouse model. The outcome of pharmacological interventions in these models remains unclear and needs further exploration [101]. Another interesting model is the gorringen minipig model, which also has limited information on pharmacological intervention studies. Liraglutide was shown to promote weight loss in these animals, in a similar method to human clinical obesity studies [102].

In an interesting study by Berg et al., it was demonstrated that administration of adiponectin improved insulin resistance in animal models [103]. Adiponectin-deficient mice developed glucose intolerance and insulin resistance. In contrast, adiponectin overexpression improved insulin sensitivity, glucose tolerance and reduced serum fatty acids.

Recent studies have demonstrated protection against obesity-induced insulin resistance through the overexpression of ER chaperones. In contrast, the knockdown of these chaperones had the opposite effect in mice [104].

## 5. Clinical Studies

Numerous clinical studies are ongoing on obesity and type 2 DM to strategize effective management and prevention. Although there are several valuable clinical studies, a few important and interesting studies are discussed here. More recently, a milestone discovery demonstrated that metabolically unhealthy obese individuals have a greater number of adipose tissue-derived exosomes compared to individuals with normal glucose tolerance and hepatic triglycerides [83].

Another interesting study demonstrated that lowering the levels of free fatty acids doubled the insulin sensitivity, increasing it from 25% to 50% of the normal values in obese individuals with type 2 DM [105]. Gastaldelli et al. elucidated that insulin resistance is directly proportional to visceral fat mass regardless of BMI [106]. In the same context, one study elucidates that body fat distribution is a crucial factor in the development of type 2 DM, independent of obesity stages. Also, individuals with insulin resistance were shown to have increased intramyocellular lipid content and decreased subcutaneous fat deposition [107]. A meta-analysis from the United States and Europe revealed that obese men have a 7-fold higher risk and obese women have a 12-fold higher chance of developing type 2 DM compared to those with normal weight [105]. The same study reported better glycemic control in type 2 DM patients on a weight loss regimen, highlighting the interdependent relationship between type 2 DM and obesity [108].

A fascinating randomized controlled trial of Semaglutide, specifically the 2-year STEP 5 trial, was conducted to test its long-term efficacy and safety in adults with obesity and type 2 DM. The participants achieved a weight loss of greater than 5% from the baseline at week 104, and also experienced improvements in diastolic blood pressure, HbA1c, fasting blood glucose, fasting serum insulin, total cholesterol, triglycerides, VLDL and LDL. This further confirms a meaningful correlation between obesity and type 2 DM as evidenced by improvements in both parameters [109]. A similar randomized clinical trial phase 3 was conducted on Tirzepatide, a glucose dependent insulinotropic polypeptide for 72 weeks on patients with obesity and type DM. The trial observed a weight reduction of 5% along with improvements in fasting insulin and lipid levels [110].

A meta-analysis on the outcome of HbA1c from lifestyle and weight loss interventions revealed that out of the 19 study groups with type 2 DM and obesity, 17 groups reported improvement in blood glucose, lipids and blood pressure over a 12-month period with a 5% reduction in weight from the baseline [111]. A significant finding was obtained in a diabetic prevention program, where moderate weight loss through lifestyle interventions in an obese population reduced the incidence of DM by 58%, whereas Metformin alone only reduced it by 31% [112].

In a retrospective study by Iglay et al., it was found that the most notable comorbid condition associated with type 2 DM is obesity (78.2%), followed by hyperlipidemia (77.2%). The highest co-prevalence was demonstrated for hypertension and hyperlipidemia (67.5%), followed by obesity and hyperlipidemia (62.5%) [113].

A study conducted by Gaich and colleagues tested the effects of fibroblast growth factor 21 (FGF21), a metabolic regulator on obese patients with type 2 DM. The study found favorable improvements in blood glucose levels, fasting insulin, body weight and adiponectin [114].

A 54-week randomized phase 2b study on Cotadutide, a GLP-1 and glucagon receptor agonist, demonstrated a significant decrease in HbA1c levels, body weight, and improvements in lipid profile and liver function tests at weeks 14 and 54 [115]. In a cross-sectional study conducted on Japanese adults, the administration of *Blautia wexlerae*, a commensal bacterium, demonstrated an inverse relationship with obesity and type 2 DM [116].

Dapagliflozin, canagliflozin and empagliflozin act by inhibiting sodium-glucose co-transporter 2 (SGLT2) in the kidney, blocking glucose reabsorption in the proximal tubule. More recent studies have reported that these drugs are effective in decreasing HbA1c levels and weight loss [117].

## 6. Management

Weight loss is the most efficient strategy for reducing the complications and comorbidities of type 2 DM. A moderate weight loss of 5 to 10% is sufficient to achieve normal blood pressure, glycemic control and increased HDL cholesterol levels [118]. Gradual weight loss leads to a decrease in adipocyte size, which in turn downregulates pathways involved in lipogenesis and oxidative stress [118]. Exercise is a key component of lifestyle interventions to achieve healthy weights and improve blood glucose levels, insulin sensitivity and lipid profiles. Most scientific guidelines recommend at least 150 min per week of moderate aerobic exercise combined with three weekly sessions of muscle strength resistance exercises. While exercise is crucial for weight loss, a combined approach of exercise and an energy-restricted diet such as a low-fat, low-carbohydrate and high-protein diet delivers better results [119]. The essential pillars in managing obesity and type 2 DM are dietary modifications and lifestyle interventions, although these can be challenging to maintain over time. Recently, different drugs have been approved to improve type 2 DM and promote weight loss such as GLP-1 receptor agonists and SGLT2 inhibitors [4].

Given the association between obesity and type 2 DM, a suitable anti-diabetic treatment for obese patients with diabetes should focus on preventing further weight gain while also using glucose-lowering agents that support weight reduction such as metformin therapy [120].

In the United States, a commonly used combination for type 2 DM and obesity management is Liraglutide with naltrexone and bupropion [121]. Metformin is the most prescribed FDA-approved medication for lowering blood sugar levels, as it increases insulin sensitivity and reduces glucose production in the liver. It also promotes weight loss and decreases food intake [122].

Bariatric surgery is highly beneficial for patients who are morbidly obese with a BMI of40 kg/m^2^ or higher or for patients with type 2 DM and a BMI of 35 kg/m^2^ or higher. This surgery effectively reduces cardiovascular events associated with morbid obesity and type 2 DM [123].

Other areas of therapeutic interest include “prebiotics”, which have been shown to improve glucose tolerance. This was demonstrated in mice on a high-fat diet that were fed prebiotics [124]. Another approach is the use of “probiotics”, which are enriched with live bacterial strains such as *Bifidobacteria* and *Lactobacilli*. These alter the gut microbiota and have been proven to be beneficial in type 2 DM by improving lipid profiles and reducing endotoxemia [125]. Several studies have reported a lower prevalence of type 2 DM and a healthy BMI in populations that consume significant amounts of polyunsaturated fatty acids, primarily found in fish [126].

Recently, intermittent fasting has been proven to be beneficial for weight reduction and improving glucose tolerance. It does this by selectively stimulating the activation of beige adipocytes in white adipose tissue (WAT) to promote “WAT browning” and by regulating the composition of intestinal microbial products, such as acetate and lactate which are known inducers of WAT browning [127].

## 7. Summary

Obesity is a significant and modifiable risk factor associated with the development and progression of type 2 DM, and the increase in obesity is the primary factor in the recent rise in the prevalence and incidence of type 2 DM.It is crucial to understand the role of obesity in the pathogenesis of type 2 DM, considering the various factors and complications associated with the condition.Obesity is a chronic progressive condition characterized by excessive and abnormal fat accumulation in the body, resulting from the consumption of more calories than the body can use, with a Body Mass Index ≥ 30 kg/m^2^.Type 2 DM is a chronic metabolic condition characterized by insulin resistance where the body is unable to effectively use insulin, leading to high blood glucose levels or hyperglycemia.Currently, over half a billion people worldwide have been diagnosed with diabetes, and this number is projected to more than double to 1.3 billion in the next 30 years.According to the World Obesity Federation 2023 atlas, it is predicted that over 51% of the global population will become overweight or obese in the next 12 years.Obesity and type 2 DM are intertwined in their pathophysiology and molecular mechanisms, influenced by various factors such as adipose tissue, homeostatic factors like adiponectin, body fat distribution, inflammation, free fatty acids, gut microbiome and dyslipidemia. Therefore, it is crucial to understand this close relationship in order to effectively manage and prevent these conditions as an urgent response to their alarming global rise.Numerous in vivo and clinical studies have highlighted the significance of a comprehensive management approach that addressed both obesity and type 2 DM simultaneously. This approach is essential for effectively handling these chronic and interconnected conditions.

## 8. Conclusions

The prevalence rates of obesity (BMI ≥ 30 kg/m^2^), extreme obesity (BMI ≥ 40 kg/m^2^) and central obesity are rising worldwide. This trend is likely to contribute to the global epidemic of type 2 DM in the coming years. Obesity is strongly believed to promote type 2 DM, with adipose tissue, liver dysfunction and skeletal muscle dysfunction playing a central role. This connection has been suggested and demonstrated by numerous significant in vivo and clinical studies. Gaining a better understanding of the relationship and causality between these factors could provide an opportunity to predict, modify and monitor the risk, as there is substantial evidence that weight loss interventions can reduce blood glucose levels. There is an urgent need for the discovery and development of multi-targeted compounds that can treat both of these conditions. Therefore, a promising opportunity for drug discovery lies in comprehensive and deeper understanding of the mechanistic links between these closely intertwined conditions.

## Figures and Tables

**Figure 1 ijms-25-01882-f001:**
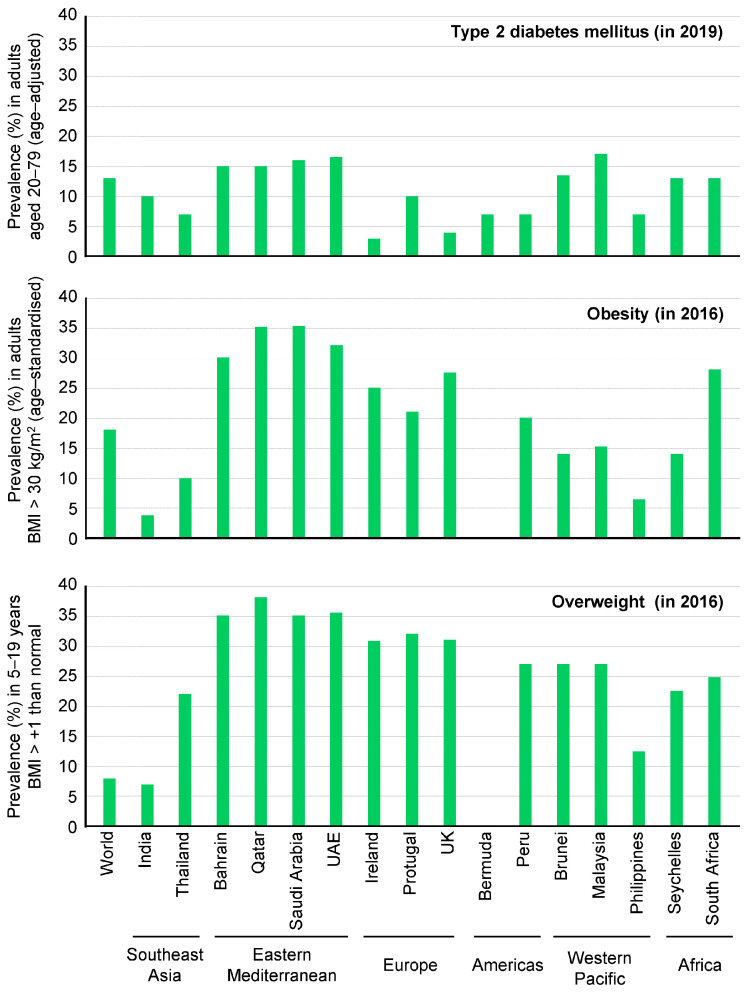
Prevalence of type 2 diabetes mellitus (in 2019), obesity (in 2016) and overweight (in 2016) in selected countries. This figure has been redrawn in a modified form from [19].

**Figure 2 ijms-25-01882-f002:**
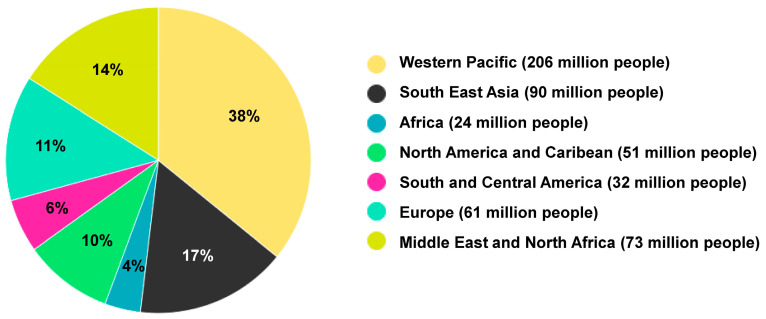
Prevalence of diabetes worldwide in 2021. A total of 573 million people suffered from diabetes in 2021. The figure has been redrawn and modified based on information from [26].

**Figure 3 ijms-25-01882-f003:**
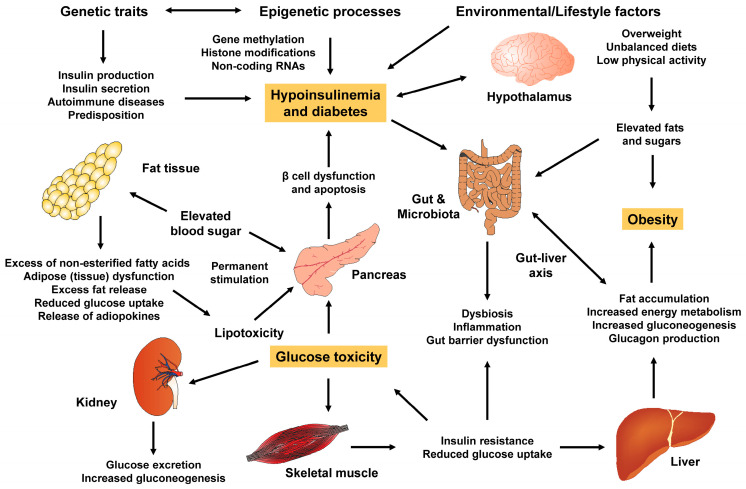
Multifactorial pathophysiology of obesity and type 2 diabetes. Genetic and epigenetic factors, along with an unhealthy lifestyle play significant roles in the development of both obesity and type 2 diabetes. Various factors, including adipokines, pro-inflammatory cytokines, non-esterified fatty acids (NEFA) and others contribute to visceral fat accumulation, β-cell dysfunction, changes in gut microbiota and gut barrier leakage. In addition, inflammatory reactions in the hypothalamus might contribute to the onset of diabetes and vice versa. Diabetes impacts energy homeostasis and hyper-activates regulatory neurons as well as the surrounding microglia in the hypothalamus.

**Figure 4 ijms-25-01882-f004:**
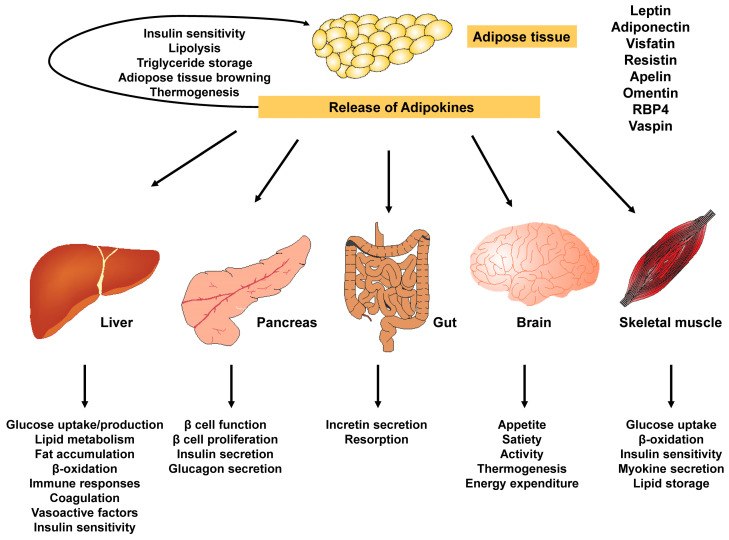
Adipokines released by adipose tissue are central in the control of endocrine and secretory functions of many organs. The adipose tissue secretes various molecules known as adipokines which act as powerful signal molecules. The activity of these adipokines impacts biological processes in liver, pancreas, gut, brain, skeletal muscles and many other organs.

**Figure 5 ijms-25-01882-f005:**
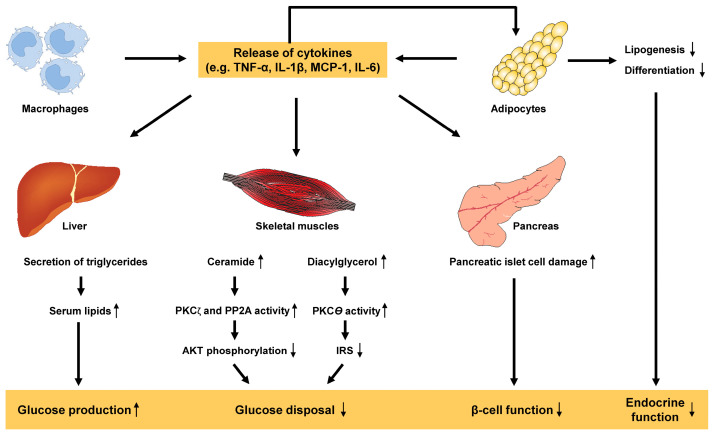
Pro-inflammatory cytokines in the pathogenesis of metabolic inflammation and insulin resistance. In the development of insulin resistance and type 2 diabetes, several cytokines such as tumor necrosis factor-α (TNF-α), interleukin-1β (IL-1β), monocyte chemoattractant protein-1 (MCP-1), IL-6 and others released by activated tissue macrophages and by adipocytes play a significant role. These pro-inflammatory molecules provoke important responses in liver, skeletal muscle, fat tissue and pancreas, resulting in endocrine dysfunction, impaired glucose disposal, impaired β-cell function and reduced suppression of glucose production. The abbreviations used are as follows: AKT, protein kinase B; IRS, insulin receptor substrate; PKCζ, protein kinase Cζ; PKC𝜃, protein kinase C𝜃; PP2A, protein phosphatase 2A.

**Figure 6 ijms-25-01882-f006:**
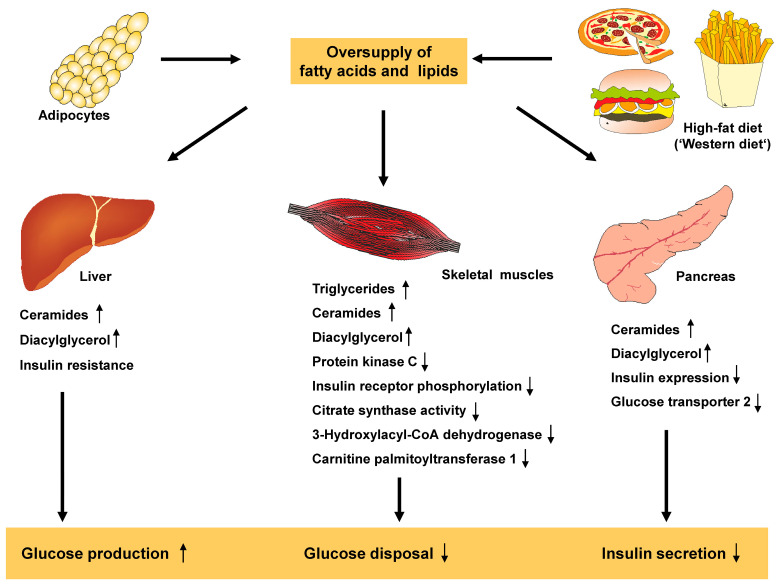
Fats and lipids in the pathogenesis of type 2 diabetes. An overabundance of fatty acids and lipids, resulting from a high-caloric diet enriched in fat (which cannot be stored in adipocytes), lead to increased levels of circulating fat that accumulate in peripheral tissues such as the liver, muscles and pancreas. This accumulation triggers numerous molecular changes that result in increased glucose production, lowered glucose disposal and impaired insulin secretion. These factors are hallmarks of diabetes. This figure was adapted in a modified form from [65].

**Table 1 ijms-25-01882-t001:** Classification of obesity based on body mass index and waist circumferences.

Condition	BMI (kg/m^2^)	Disease Risk Relative to Normal Weight and Waist Circumference Men ≤ 40 inches (≤102 cm) Women ≤ 35 inches (≤88 cm)
Normal	18.5–24.9	data
Overweight	25.0–29.9	Increased
Obese	30.0–34.9 (class 1)	High
	35.0–39.9 (class 2)	Very high
Extremely Obese	≥40	Extremely high

**Table 2 ijms-25-01882-t002:** Anthropometric assessment of body fat percentage as a measure of adiposity.

Condition	Males	Females
Essential fat	<15%	<10%
Athletes	15–19%	10–14%
Fit	20–24%	15–19%
Acceptable	25–29%	20–24%
Pre-obesity	30–34%	25–29%
Obesity	>35%	>30%

## Data Availability

This review only presents data that were previously published. No new data was generated.

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
