# Peer review of "The Role of Obesity in Type 2 Diabetes Mellitus—An Overview"

_ijms, 2024, doi:10.3390/ijms25031882_

Round 1
Reviewer 1 Report
Comments and Suggestions for Authors
The review "The Role of Obesity in Type 2 Diabetes Mellitus – An Overview" by Chandrasekaran and Weiskirchen is a well written and succinct review of the relationship between obesity and diabetes. The current statistics on obesity and T2D at the beginning are very useful. It addresses several aspects of the metabolic dysfunction of obesity and diabetes at the mechanistic level and population level with fairly surface level coverage. It focuses most attention on the prevalence differences between different ethnicities/countries, and the role of adipose tissue in driving the mechanisms of T2D development.
I think that discussions highlighting that the location of extra-adipose fat plays a large role in T2D development rather than overall adiposity/weight (either in the beginning of section 2 or in section 3.6) could benefit from mentioning that hepatic lipid accumulation, in particular, is a critical driver of T2D. In humans the degree of fat in the liver correlates with the severity of systemic insulin resistance, and obese humans and rodent models that don't have excess hepatic lipids are metabolically healthier despite being obese.
The mechanistic sections focus on adipose tissue derived mechanisms, which makes sense given the review is highlighting the role of obesity in T2D, but the short section on beta-cell dysfunction and the gut microbiome seem out of place. There are many other considerations for T2D mechanism including metabolic dysfunction of gut hormones, liver, muscle, and the brain. So it feels like the authors picked a few areas to talk about at random. I would recommend getting rid of the paragraphs about the pancreas and microbiome and focusing the mechanistic explanations to adipose tissue, and also highlight that you are choosing the focus on adipose tissue while acknowledging that many other organs play roles in the pathogenesis of T2D.
The 'in vivo studies' section I would rename 'in vivo models' because you are really listing obesity and T2D models with one or two studies that have been done in those models, rather than a thorough discussion of rodent work that demonstrates the mechanisms you laid out above.
Minor: line 396 A is capitalized.
Author Response
Dear Reviewer 1,
many thanks for your valuable comments. Please find our response to your suggestions in the attached pdf-file.
Regards
Ralf Weiskirchen

Reviewer 2 Report
Comments and Suggestions for Authors
Comments to the author:
The Manuscript " The Role of Obesity in Type 2 Diabetes Mellitus – An Overview. The manuscript suggest that the recent management approaches for the treatment of these associated conditions. Additionally, in 24 vivo studies on obesity and type 2 DM are discussed here as they pave the way for more rigorous 25 development of therapeutic approaches.
Revisions required:
1- Kindly elaborate the role of white adipose tissue and brown adipose tissue. Particularly browning of adipose tissue
2- Please explain the role of specifics genes and proteins responsible for healthy Adipose tissue ie. PgC1 alpha, Ho1.
Comments on the Quality of English LanguageMinor corrections are required
Author Response
Dear Reviewer 2,
many thanks for your valuable comments. Please find our response to your suggestions in the attached pdf-file.
Regards
Ralf Weiskirchen
